# TNF-α-Mediated RIPK1 Pathway Participates in the Development of Trigeminal Neuropathic Pain in Rats

**DOI:** 10.3390/ijms23010506

**Published:** 2022-01-03

**Authors:** Jo Young Son, Jin Sook Ju, Yu Mi Kim, Dong Kuk Ahn

**Affiliations:** Department of Oral Physiology, School of Dentistry, Kyungpook National University, 2177 Dalgubeol-daero, Chung-gu, Daegu 41940, Korea; n-violetjy@nate.com (J.Y.S.); jsju@knu.ac.kr (J.S.J.); boboatom@naver.com (Y.M.K.)

**Keywords:** RIPK1, inferior alveolar nerve injury, mechanical allodynia, TNF-α, trigeminal neuropathic pain

## Abstract

Receptor-interacting serine/threonine-protein kinase 1 (RIPK1) participates in the regulation of cellular stress and inflammatory responses, but its function in neuropathic pain remains poorly understood. This study evaluated the role of RIPK1 in neuropathic pain following inferior alveolar nerve injury. We developed a model using malpositioned dental implants in male Sprague Dawley rats. This model resulted in significant mechanical allodynia and upregulated RIPK1 expression in the trigeminal subnucleus caudalis (TSC). The intracisternal administration of Necrosatin-1 (Nec-1), an RIPK1 inhibitor, blocked the mechanical allodynia produced by inferior alveolar nerve injury The intracisternal administration of recombinant rat tumor necrosis factor-α (rrTNF-α) protein in naive rats produced mechanical allodynia and upregulated RIPK1 expression in the TSC. Moreover, an intracisternal pretreatment with Nec-1 inhibited the mechanical allodynia produced by rrTNF-α protein. Nerve injury caused elevated TNF-α concentration in the TSC and a TNF-α block had anti-allodynic effects, thereby attenuating RIPK1 expression in the TSC. Finally, double immunofluorescence analyses revealed the colocalization of TNF receptor and RIPK1 with astrocytes. Hence, we have identified that astroglial RIPK1, activated by the TNF-α pathway, is a central driver of neuropathic pain and that the TNF-α-mediated RIPK1 pathway is a potential therapeutic target for reducing neuropathic pain following nerve injury.

## 1. Introduction

The characterization of neuropathic pain is continuous and/or paroxysmal pain associated with dysesthesia or allodynia. It has been previously established that peripheral neuropathies caused by lesions in the peripheral nervous system are extremely difficult to treat as multiple underlying mechanisms can be involved and effective interventions are dependent on the cause. Drug development for treating neuropathic pain often focuses therefore on the inhibition or relief of certain underlying causes or symptoms.

Receptor-interacting serine/threonine protein kinase 1 (RIPK1), a member of the serine-threonine protein kinase family, is a well-known and critical mediator of necroptosis and a regulator of the responses to cellular stress and inflammation [1,2,3,4]. Previous reports have demonstrated the association of RIPK1 found with several disorders, such as inflammatory disease, ischemic injury, axonal degeneration, autoimmune disease, and different cancers [5]. Recent evidence supports the involvement of RIPK1 in the processes underlying nociceptive information. RIPK1 expression significantly increases at the site of injury following a spinal cord laminectomy [6], a spinal cord injury [7], and in a chronic constriction injury model [8,9]. Moreover, a blockade of RIPK1 expression alleviates hyperalgesia and mechanical allodynia in rats with sciatic nerve injury [9]. RIPK1 has therefore generated interest as a potential target for neuropathic pain treatment. However, its function in trigeminal neuropathic pain remains unknown, with previous findings suggesting that it can modulate pain processing after various types of nerve injury.

Tumor necrosis factor (TNF)-α is regarded as a proinflammatory cytokine with a central function in regulating pain communication between the immune system and the brain [10]. Generally, TNF-α, mediated through the glial system, plays a central role in neuropathic pain. TNF-α is produced by spinal glial cells (both microglia and astrocytes) following nerve injury or inflammation [11,12]. Although these prior findings suggest a central role for TNF-α in the development of neuropathic pain [13,14], the precise mechanisms by which TNF-α affects the RIPK1 pathway are poorly understood.

We hypothesized that the TNF-α-mediated activation of the RIPK1 pathway plays a key role in trigeminal neuropathic pain following inferior alveolar nerve injury. For this purpose, we examined changes in RIPK1 expression and TNF-α concentration in the trigeminal subnucleus caudalis (TSC) following inferior alveolar nerve injury in a rat model. We also used this model to examine the changes in the air-puff thresholds and TNF-α concentration after blockade of the RIPK1 pathway by intracisternally administered Nec-1, an RIPK1 inhibitor. Finally, we evaluated the cellular localization of RIPK1 and TNF receptor in the TSC of these animals by double immunofluorescence analysis.

## 2. Results

### 2.1. Inferior Alveolar Nerve Injury Produces Mechanical Allodynia and Upregulated RIPK1 Expression

Figure 1 illustrates the detected alterations in air-puff thresholds and RIPK1 expression in the TSC following inferior alveolar nerve injury in an animal model of trigeminal neuropathic pain that was developed in male SD rats via the malpositioning of dental implants. A significant decrease was observed in the air-puff thresholds ipsilateral to the nerve injury model (*p* < 0.05, F_(2.17)_ = 800.124, Figure 1A). Mechanical allodynia induced by the inferior alveolar nerve injury was observed on POD 1 and persisted until POD 40. Although a sham group also showed decreased air-puff thresholds, this was not statistically significant. The inferior alveolar nerve injury group also produced a significantly increased RIPK1 expression in the TSC on POD 3 and 7 (*p* < 0.05, Figure 1B). On POD 50, RIPK1 expression was recovered to basal value levels. The sham-operated control animals showed comparable RIPK1 expression to naïve rats.

### 2.2. Effects of an RIPK1 Inhibitor on Mechanical Allodynia

Figure 2 presents the observed alterations in the air-puff thresholds subsequent to the blockade of the RIPK1 pathway on POD 3, 7, and 21, respectively, in the rat model. A single intracisternal injection of Nec-1 (1 or 10 µg), an RIPK1 inhibitor, had significant anti-allodynic effects in a dose-dependent manner on POD 3 (F_(2.18)_ = 44.791, *p* < 0.05, Figure 2A). An anti-allodynic effect was evident at 4 h after intracisternal administration of Nec-1 (10 μg) and had disappeared by 24 h post-injection. Additionally, a low dose of Nec-1 (1μg) induced an anti-allodynic effect within 5 h of its intracisternal administration and again had disappeared by 24 h post-injection. On POD 7, both doses of Nec-1 (1 or 10 μg) produced prolonged anti-allodynic effects (F_(2.19)_ = 53.625, *p* < 0.05, Figure 2B). On the other hand, only the higher dose of Nec-1 (10 μg) produced anti-allodynic effects on POD 21 (F_(2.19)_ = 13.4, *p* < 0.05, Figure 2C) although at a lower magnitude than those observed at POD 3 or 7. Intracisternal administration of vehicle did not affect air-puff thresholds.

### 2.3. Effects of an rrTNF-α Protein on Air-Puff Thresholds and RIPK1 Expression in Naïve Rats

Alterations in the air-puff thresholds and RIPK1 expression in naïve rats following the intracisternal administration of rrTNF-α protein are illustrated in Figure 3. A single intracisternal injection of the rrTNF-α protein (at either 20 or 200 ng) produced significant decreases in the air-puff threshold when compared to the vehicle-treated group (F_(2.18)_ = 172.249, *p* < 0.05, Figure 3A). The lower 20 ng dose of rrTNF-α protein also produced mechanical allodynia within 30 min of the intracisternal administration and the allodynic effects persisted for 4 h and abated within 24 h. The higher 200 ng dose of rrTNF-α protein also produced mechanical allodynia within 30 min after the intracisternal administration which persisted for over 6 h and then also abated within 24 h. Analysis by Western blotting revealed significant upregulation of RIPK1 expression in the TSC at 2 h after intracisternal administration of the high-dose rrTNF-α protein (*p* < 0.05, Figure 3B). The vehicle control did not affect the air-puff thresholds and RIPK1 expression. To then investigate the possible participation of the RIPK1 pathway in this rrTNF-α-protein-induced mechanical allodynia, we evaluated these effects following an intracisternal pretreatment with Nec-1 at 4 h before the rrTNF-α protein injection in naïve rats. Whereas vehicle administration did not affect the mechanical allodynia in the naïve animals, pretreatment with Nec-1 completely blocked the development of mechanical allodynia produced by the intracisternal administration of rrTNF-α protein (Figure 3C).

### 2.4. TNF-α-mediated RIPK1 Pathway Participates in Trigeminal Neuropathic Pain

The TNF-α concentration was found to increase in a time-dependent manner, coincident with the development of mechanical allodynia in the TSC, following the placement of malpositioned dental implants in the rat model (Figure 4A). The sham group did not show any alteration in the TNF-α concentration compared with the naïve group. However, ELISA analysis established that the inferior alveolar nerve injury increased the TNF-α concentration compared with the sham group on POD 1, 3, and 5 (*p* < 0.05). As shown in Figure 4B, the intracisternal administration of a TNF-α antibody (2 or 20 μg) produced anti-allodynic effects (*p* < 0.05, F_(2.18)_ = 48.416), which appeared 2 h after the 20 μg injection and 4 h after the 2 μg injection. These effects had dissipated by 24 h post-injection. Vehicle administration did not alter the mechanical allodynia induced by the malpositioned dental implantation. Western blotting indicated that the intracisternal administration of TNF-α antibody, but not vehicle, attenuated the upregulated RIPK1 expression in the TSC, resulting from the inferior alveolar nerve injury on POD 3 (*p* < 0.05, Figure 4C).

### 2.5. Colocalization of RIPK1 and TNFR1 in the TSC

Figure 5 presents the results of double immunofluorescence staining for RIPK1 and TNFR1 and their cellular localization in the TSC on POD 3. The colocalization of RIPK1 (Figure 5A) and TNFR1 (Figure 5B) was observed with GFAP, an astrocyte marker, respectively.

## 3. Discussion

Our present study first demonstrated that TNF-α-mediated activation of RIPK1 pathway in astrocytes contributes to the development of trigeminal neuropathic pain. Inferior alveolar nerve injury in a rat model induced significant mechanical allodynia and increased both TNF-α concentration and RIPK1 expression. The intracisternal administration of Nec-1, an RIPK1 inhibitor, attenuated the trigeminal neuropathic pain. Moreover, the intracisternal administration of TNF-α antibody attenuated both the trigeminal neuropathic pain and the upregulated RIPK1 expression in the TSC. Finally, double immunofluorescence analyses revealed the colocalization of RIPK1 and TNFR1 with an astrocytes marker. These results indicated that the neuropathic pain had been mediated through TNF-α-mediated activation of the RIPK1 pathway in astrocytes. Hence, a blockade of the astroglial TNF-α-mediated RIPK1 pathway induced a hypothetically effective treatment strategy for trigeminal neuropathic pain.

Our present study also revealed that an inferior alveolar nerve injury produces prolonged mechanical allodynia and upregulated RIPK1 expression in the TSC. This is consistent with previous studies reporting that animals with high levels of RIPK1 exhibited significant allodynia in behavioral tests [9]. The present study also demonstrated that a blockade of the RIPK1 pathway through the intracisternal administration of Nec-1 significantly blocked trigeminal neuropathic pain in the experimental rats. RIPK1 contributed not only early (POD 3) but also late (POD 21) to the allodynic effects induced by the inferior alveolar nerve injury. These findings suggest that the RIPK1 pathway plays a pivotal role in the onset of trigeminal neuropathic pain following inferior alveolar nerve injury and thus RIPK1 functions as an important mediator in the development of neuropathic pain. Our results are in line with previous findings suggesting that upregulated RIPK1 expression may provide an essential mechanism for the pathogenesis of neuropathic pain. Increased RIPK1 expression has been observed in various types of neuropathic pain in rats harboring a spinal cord injury [6,7] or sciatic nerve injury [8,9]. Moreover, the intraperitoneal administration of Nec-1 was reported to ameliorate both mechanical allodynia and hyperalgesia in a rat model of sciatic nerve injury [9]. Although these results indicated the participation of the RIPK1 pathway in the initiation of neuropathic pain, little is yet known about the precise cellular mechanisms involving this pathway in pain processing in the orofacial area.

TNF-α has been shown to be directly involved in pain responses in several animal models of peripheral nerve injury. Intrathecal injections of TNF-α are reported to elicit neuropathic pain-like behavior [15], and produce neuropathic hyperalgesia in partial sciatic transection models [16,17]. Moreover, the intrathecal or intravenous injection of TNF-α inhibitor (etanercept) reduces neuropathic pain behaviors in diabetic mice [18] and also the pain responses to spinal cord injury in the rat [19]. Furthermore, transgenic mice that overexpress TNF display exaggerated mechanical hypersensitivity compared to their wild type counterparts following peripheral nerve injury [20]. It is widely accepted that TNF-α is produced during the initiation of an inflammatory cascade, and contributes to neuropathic pain [21,22]. Notably, few studies have confirmed the upregulation of TNF-α expression within the spinal cord following nerve injury. Our present data have revealed that an inferior alveolar nerve injury upregulates TNF-α concentration in TSC and that an intracisternal injection of TNF-α antibody abolishes mechanical allodynia, a hallmark of neuropathic pain. These results indicate that central TNF-α is involved in the development of neuropathic pain.

We investigated the role of RIPK1 in the processing of TNF-α signaling and found that an intracisternal administration of rrTNF-α protein produced significant mechanical allodynia in the orofacial area and upregulated RIPK1 expression in the TSC in naïve rats. A blockade of RIPK1 pathway by pretreatment with Nec-1 significantly inhibited this intracisternally administered rrTNF-α-protein-induced mechanical allodynia in naïve rats. However the intracisternal administration of Nec-1 did not induce mechanical allodynia in naïve rats (data not shown). These findings indicate that the RIPK1 pathway is activated by the actions of TNF-α in the TSC. This TNF-α-mediated RIPK1 pathway has been studied by the cell death cellular signaling mechanism. In TNF-α-stimulated cells, RIPK1 was reported to be activated intracellularly through the TNF receptor [23] and to regulate the events downstream of the TNF receptor [24,25]. Moreover, our present study demonstrated that a blockade of TNF-α via an intracisternally administered TNF-α antibody attenuates RIPK1 expression in the TSC in rats with an inferior alveolar nerve injury. Taken together with previous data, our current results suggest that TNF-α-mediated activation of the RIPK1 pathway is involved in the initiation of neuropathic pain following inferior alveolar nerve injury.

TNF-α has two receptors and both TNFR1 and 2 are known to be involved in neuropathic pain [26,27]. TNFR1 recruits RIPK1, a member of the membrane-bound receptor signaling complex that activates the pro-survival gene. We attempted in our experiments to confirm the location of TNFR1 as it activates RIPK1 signaling [28]. Based on our double immunofluorescent staining data, RIPK1 and TNFR1 are principally located in astrocytes (i.e., not neurons or microglia), so we conclude from this that an astroglial TNF-α-mediated RIPK1 pathway plays an important role in the development of neuropathic pain following inferior alveolar nerve injury.

## 4. Materials and Methods

### 4.1. Animals

The animal experiment followed the protocols of the Committee of the School of Dentistry, Kyungpook National University (No. 20210034). Adult male Sprague Dawley (SD) rats were used. A total of 269 SD rats were purchased from Jung-Ang experimental laboratory (Seoul, Korea). All animals, weighting between 200 and 220 g, were maintained in a pathogen-free environment, in a 12:12 h light-dark cycle, and maintained at 23 ± 1 °C. Food and water were provided ad libitum. All experiments followed the ethical guidelines set for the investigation of experimental pain in conscious animals by the International Association for the Study of Pain (IASP). Each rat was used only once and all rats were handled at least 5 days before the surgery or behavioral testing to minimize stress. All of these experiments were conducted in a blinded fashion by a single researcher and the animals were randomized during the experiments.

### 4.2. Trigeminal Neuropathic Pain Animal Model

In the trigeminal neuropathic pain model, SD rats were anesthetized with a ketamine/xylazine (40/4 mg/kg, intramuscular; i.m.) mixture. After anesthetization, the second molar of the left mandibular was extracted and a mini-dental implant was placed (diameter, 1 mm; length, 4 mm; donated by Megagen, Daegu, Korea) to injure the inferior alveolar nerve, as previously described [29]. Similarly, a sham group underwent surgery and extraction but no dental implantation. A control group of naive rats who did not undergo the operation were also used. For the final analyses, we used only data derived from animals who demonstrated inferior alveolar nerve injury caused by the malpositioned dental implants.

### 4.3. Intracisternal Catheterization

While under anesthesia, intracisternal administrations were performed in the rats using a stereotaxic frame, and a polyethylene tube 10 (PE10) was implanted into each animal, as previously described [30,31,32,33]. A small hole was generated in the atlantooccipital membrane and dura using a 27-gauge needle. Using this prepared hole, a PE10 was inserted into the intracisternal area, with the cannula tip dorsal to the obex. Positioned subcutaneously to the skull, a polyethylene tube was then secured in place using dental acrylic resin and a stainless steel screw. Animals were allowed to recover for 3 days after surgery, a period previously demonstrated to be sufficient [34,35]. Following intracisternal catheterization, any animal demonstrating motor dysfunction or malpositioning of the catheter were excluded from further analysis.

### 4.4. Evaluation of Mechanical Allodynia

We measured mechanical allodynia using previously described methods [36,37,38]. For the behavioral observations, we placed each individual animal into a customized cage, in a noise-free and darkened room. Animals were allowed to acclimate for a minimum of 30 min before behavioral testing. Withdrawal behavior was determined using 10 stimulations at a constant air-puff pressure (4 s duration and 10 s intervals) delivered ipsilaterally to the side of the nerve injury. The intensity and intervals of the air-puff pressure were controlled using a pico-injector (Harvard Apparatus, Holliston, MA). Following the establishment of injury to the inferior alveolar nerve, the most sensitive area was determined using air-puff stimulation, as previously described [30] and was found to be the lower jaw and the mouth angle area of the facial region. Using a 26-gauge metal tube of length 10 cm, these sensitive areas were subjected to air-puff stimulation and maintained at a 90° angle at 1 cm from the skin. An air-puff threshold was considered to be the pressure at which an individual rat responded to 50% of the trials. The maximum air-puff stimulation cut-off was 40 psi, as previously described [35,39,40]. Air-puff stimulations were performed during the day time only (08:00–18:00 h). The naive animals only responded to pressures higher than 40 psi.

### 4.5. Immunofluorescence Staining

For immunofluorescence analysis, the rats were sacrificed and perfused transcardially with 0.9% saline, followed by 4% paraformaldehyde in 0.1 M phosphate buffer (PB, pH 7.4). The caudal medulla was dissected and a portion of this tissue then underwent additional fixation using the same fixative solution for 2 h at 4 °C. Following 2 h fixation, the sample was then placed in 30% sucrose in 0.1 M PB solution overnight. Prior to processing for immunofluorescence analysis, transverse frozen sections (free-floating, 16 µm) were cut using a cryostat. All tissue was blocked with a 10% normal donkey serum in phosphate buffered saline (PBS, pH 7.4) for 30 min at room temperature and for the double immunofluorescence, the sections were incubated overnight at 4 °C with a mixed solution of rabbit anti-RIPK1 (1:100, Novus, Centennial, CO, USA) or a rabbit anti-TNFR1 (1:100, Antibody-online, Wilton, UK) with a mouse anti-NeuN (neuronal marker; 1:5000; Millipore, Burlington, MA, USA), a goat anti-Iba-1 (microglial marker; 1:10,000; Abcam, Cambridge, MA, USA), or mouse anti-GFAP (astrocytic marker; 1:10,000; Cell Signaling Technology, Danvers, MA, USA). Following these primary antibody incubations, samples were than incubated with a mixture of anti-rabbit Cy3 (1:200; Jackson ImmunoResearch) and anti-mouse or anti-goat FITC (1:200; Jackson ImmunoResearch). Images were collected with a fluorescence microscope (BX 631 and U-RFL-T; Olympus, Japan) or a confocal laser scanning microscope (LSM 510; Carl Zeiss, Jena, Germany) to observe the immunofluorescence signals.

### 4.6. Western Blotting

For Western blotting analysis, the ipsilateral caudal medulla was collected and homogenized as described previously by Yoon et al. [41]. Total proteins were then extracted from the homogenized samples using lysis buffer. Equal amounts of protein samples (30 μg) were subjected to 10% NuPAGE Novex Bis-Tris gel (Invitrogen) electrophoresis, and then transferred onto nitrocellulose membranes. Membranes were subsequently incubated overnight at 4 °C with anti-RIPK1 antibody (1:2000; Rockland, Limerick, PA, USA) and then with a secondary anti-rabbit IgG antibody (1:5000; Bio-Rad, Hercules, CA, USA). Finally, the band signals were visualized via enhanced chemiluminescence (ECL) kit (Amersham Imager 600; GE Healthcare, Piscataway, NJ, USA). The Image J analysis system (NIH, Bethesda, MD, USA) was used for the quantification of specific bands. GAPDH antibodies (1:10,000; Santa Cruz Biotechnology, Santa Cruz, CA, USA) were used as a loading control.

### 4.7. Enzyme-Linked Immunosorbent Assay (ELISA) 

At the indicated time points, protein supernatants were collected and evaluated for cytokine levels using commercial ELISA kits, in accordance with the manufacturer’s protocol. The TNF-α ELISA kit was purchased from R&D Systems (Minneapolis, MN, USA) and the optical density (OD) was measured spectrophotometrically at wavelengths of 420–570 nm. The data were expressed relative to a standard curve prepared for TNF-α.

### 4.8. Chemicals

Nec-1 was purchased from Selleckchem (Houston, TX) and dissolved in a combination of 45% PEG 300, 5% DMSO, and 50% DDW. TNF-α antibody (R&D systems) was dissolved in normal saline. Recombinant rat TNF-α (rrTNF-α) protein was obtained from R&D systems and dissolved in sterile PBS.

### 4.9. Experimental Protocols

#### 4.9.1. Participation of RIPK1 in Trigeminal Neuropathic Pain

Prolonged nociceptive behavior was observed following inferior alveolar nerve injury in several previous studies [14,38]. Here, we also evaluated how the ipsilateral air-puff threshold changes after inferior alveolar nerve injury produced by malpositioned dental implants (*n* = 7 per group). Mechanical allodynia was determined at 2 days prior to the operation and at 1, 2, 3, 5, 7, 10, 14, 21, 30, 35, 40, 45, and 50 days after the inflicted injury to the nerve. Western blotting analysis was conducted to evaluate RIPK1 expression following inferior alveolar nerve injury (*n* = 6 per group) on POD 3, 7, 21, and 50. Neuropathic mechanical allodynia was assessed after a blockade of RIPK1. The dose of Nec-1, an RIPK1 inhibitor, was based on prior reference [42]. Nec-1 (1 or 10 µg/10µL) was administered intracisternally using the implanted PE10 tube in the experimental rats with alveolar nerve injury on POD 3, 7, and 21. Mechanical allodynia was measured at 0, 0.5, 1, 1.5, 2, 2.5, 3, 4, 5, 6, 7, 8, and 24 h following the intracisternal administration of Nec-1 or vehicle (*n* = 7 per group).

#### 4.9.2. Effects of Intracisternally Administered rrTNF-α Protein on Air-Puff Thresholds and RIPK1 Expression in Naïve Rats

To evaluate the effects of rrTNF-α protein on air-puff thresholds, the rrTNF-α protein (20 or 200 ng/10 µL) was injected intracisternally into naïve rats. Mechanical allodynia was then measured at 0, 0.5, 1, 1.5, 2, 2.5, 3, 4, 5, 6, and 24 h after treatment with the rrTNF-α protein or vehicle (*n* = 7 per group). RIPK1 protein expression changes in the TSC were determined by Western blot analysis 2 h after treatment with rrTNF-α protein (*n* = 8 per group). To assess the participation of RIPK1 in TNF-α-induced mechanical allodynia, the effects of Nec-1, an RIPK1 inhibitor, on mechanical allodynia generated by the administration of rrTNF-α protein were monitored in naïve rats. Nec-1 (10 µg/10 µL) was administered 4 h before the rrTNF-α protein (200 ng/10 µL) injection and mechanical allodynia was evaluated at 0, 0.5, 1, 1.5, 2, 2.5, 3, 4, 5, 6, and 24 h after rrTNF-α protein injection (*n* = 7 per group).

#### 4.9.3. Participation of the TNF-α-Mediated RIPK1 Pathway in Trigeminal Neuropathic Pain

ELISA analysis was performed (*n* = 10 per group) on POD 1, 3, and 5 to evaluate changes in the TNF-α concentration subsequent to inferior alveolar nerve injury. We also investigated changes in neuropathic mechanical allodynia and RIPK1 expression after a blockade of TNF-α in the rats with inferior alveolar nerve injury. On POD 3, TNF-α antibody (2 or 20 µg/10 µL) was administered intracisternally followed by saline flushing. Mechanical allodynia was then measured at 0, 0.5, 1, 1.5, 2, 2.5, 3, 4, 5, 6, 7, 8, and 24 h after treatment with TNF-α antibody or vehicle (*n* = 7 per group). Western blotting was used to evaluate expression changes in RIPK1 within the TSC at 6 h after treatment with TNF-α antibody in the rats with inferior alveolar nerve injury (*n* = 8 per group).

#### 4.9.4. Co-localization of RIPK1 and TNFR1 in the TSC

Double immunofluorescence staining (*n* = 6 per group) was completed on POD 3 for RIPK1 and TNFR1 to evaluate their subcellular localization within the ipsilateral TSC, using markers for neurons (NeuN), microglia (IBA1), or astrocytes (GFAP).

### 4.10. Statistical Analysis

Repeated measures analysis of variance (ANOVA), followed by the Holm–Sidak post hoc test, was utilized to statistically analyze all behavioral data. For qualitative data obtained by Western blotting or ELISA, changes were evaluated using a Student’s *t*-test for two group comparisons, and one-way ANOVA followed by Holm–Sidak post hoc analysis for multigroup comparisons. For statistical comparisons, a *p* value of < 0.05 was considered to indicate significance. All data are presented as the mean ± standard error of the mean (SEM).

## 5. Conclusions

Our results provide a connection between trigeminal neuropathic pain and TNF-α-RIPK1 pathway. Inferior alveolar nerve injury results in a significant upregulation of RIPK1 expression in the TSC and a blockade of the RIPK1 pathway effectively relieves mechanical allodynia. A single intracisternal administration of TNF-α antibody attenuates trigeminal neuropathic pain and suppresses the upregulation of RIPK1 expression. Double immunofluorescence analyses revealed the colocalization of RIPK1 and TNFR1 with astrocyte markers. Hence, an astroglial TNF-α-mediated RIPK1 pathway is involved in important pathogenic mechanisms underlying our rat pain model. Modulation of this pathway may be a viable therapeutic strategy to alleviate neuropathic pain.

## Figures and Tables

**Figure 1 ijms-23-00506-f001:**
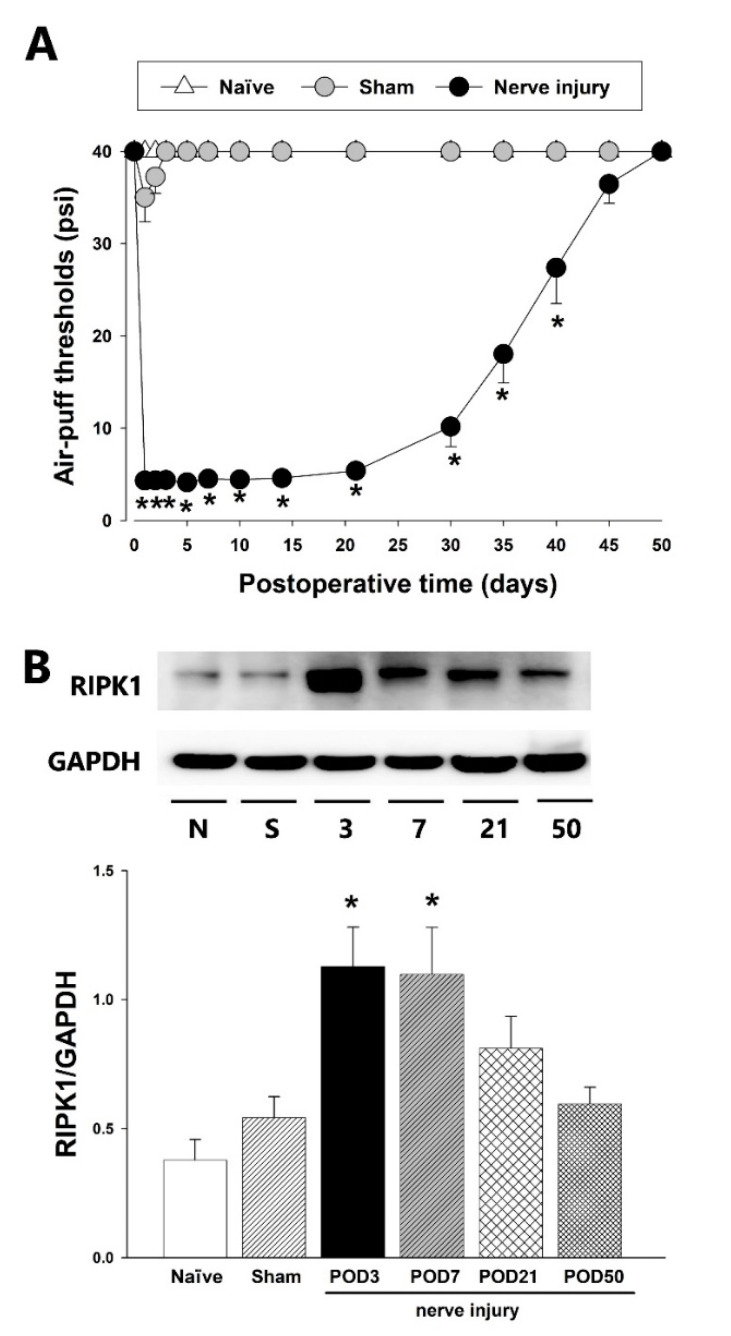
Changes in the air-puff thresholds and RIPK1 expression following inferior alveolar nerve injury. (**A**) Significant mechanical allodynia was induced by the inferior alveolar nerve injury compared with the sham-operated group. A cut-off pressure was determined as naïve animals did not respond to pressures below 40 psi (repeated measure ANOVA with Holm–Sidak post hoc tests, *n* = 7). (**B**) Western blotting analysis showed upregulation of RIPK1 expression on POD 3, 7 in the ipsilateral dorsal parts of the caudal medulla. GAPDH was used as a loading control (one-way ANOVA with Holm–Sidak post hoc tests, *n* = 8). All mean ± SEM, * *p* < 0.05, sham vs. alveolar nerve injury group.

**Figure 2 ijms-23-00506-f002:**
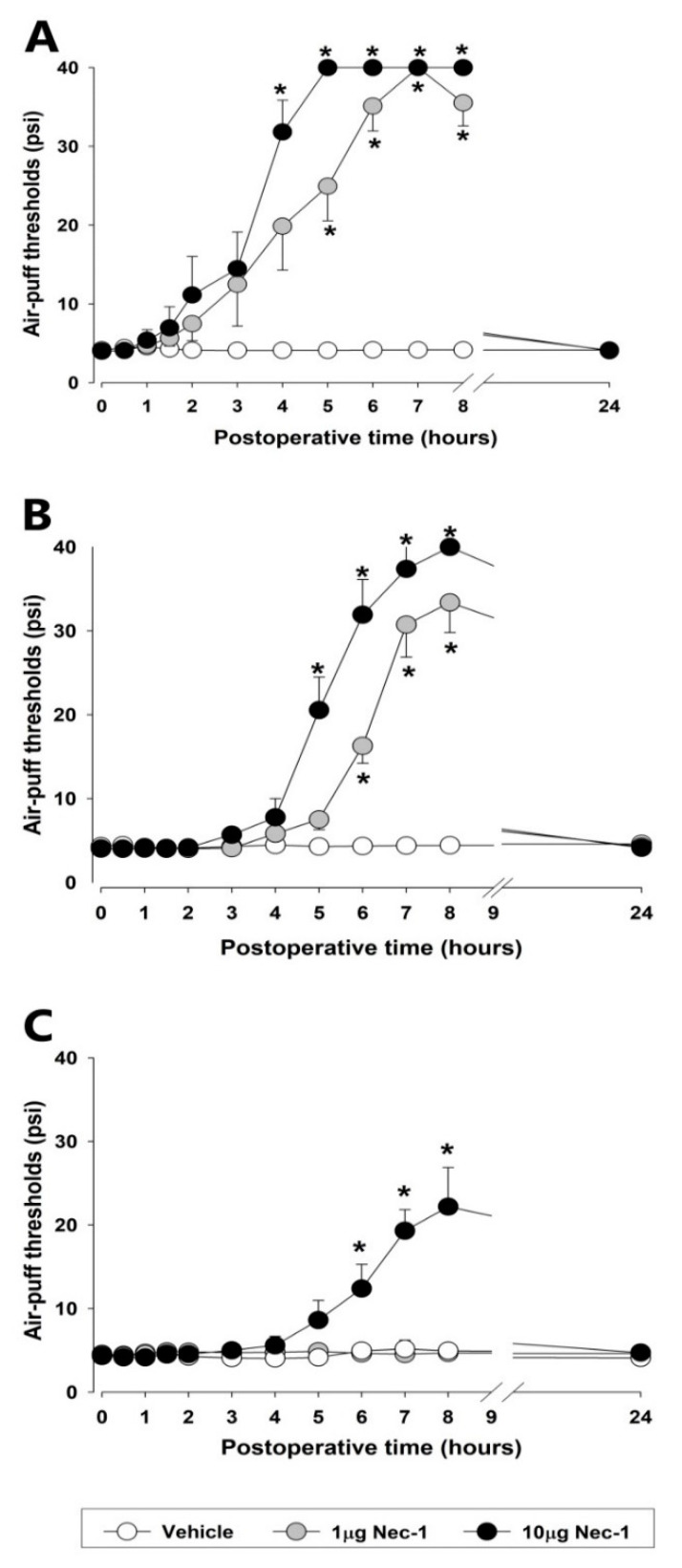
The effects of early or late treatment with Necrosatin-1 (Nec-1), an RIPK1 inhibitor, on mechanical allodynia in rats with an inferior alveolar nerve injury. (**A**–**C**) The intracisternal administration of Nec-1 (1 or 10 μg) produced anti-allodynic effects compared to the vehicle. The treatment with Nec-1 significantly attenuated mechanical allodynia on POD 3, 7, and 21 (repeated measure ANOVA with Holm–Sidak post hoc tests, *n* = 7). All mean ± SEM, * *p* < 0.05, vehicle vs. Nec-1-treated group.

**Figure 3 ijms-23-00506-f003:**
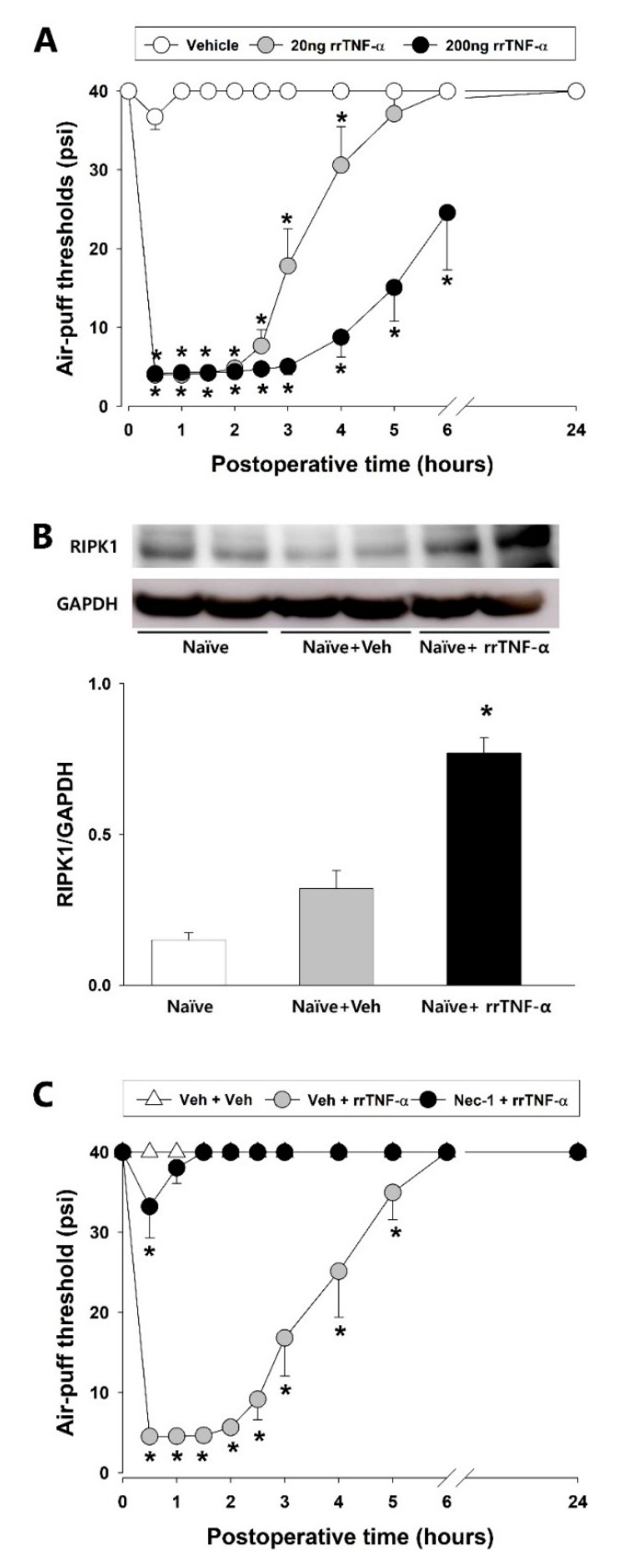
Intracisternal treatments with rrTNF-α protein produced mechanical allodynia and upregulated RIPK1 expression in naïve rats. (**A**) The intracisternal administration of rrTNF-α protein (20 ng or 200 ng) produced significant decreases in the air-puff thresholds, as compared to the vehicle-treated group (repeated measure ANOVA with Holm–Sidak post hoc tests, *n* = 7). (**B**) Western blot analysis revealed that RIPK1 expression was upregulated in ipsilateral dorsal parts of the caudal medulla at 2 h after an intracisternal treatment with 200 ng rrTNF-α protein (one-way ANOVA with Holm–Sidak post hoc tests, *n* = 8). (**C**) Nec-1 pretreatment prevented the rrTNF-α-protein-induced increases in the air-puff thresholds compared to the vehicle-treated group (repeated measure ANOVA with Holm–Sidak post-hoc tests, *n* = 7). All mean ± SEM, * *p* < 0.05, vehicle vs. rrTNF-α-protein-treated group.

**Figure 4 ijms-23-00506-f004:**
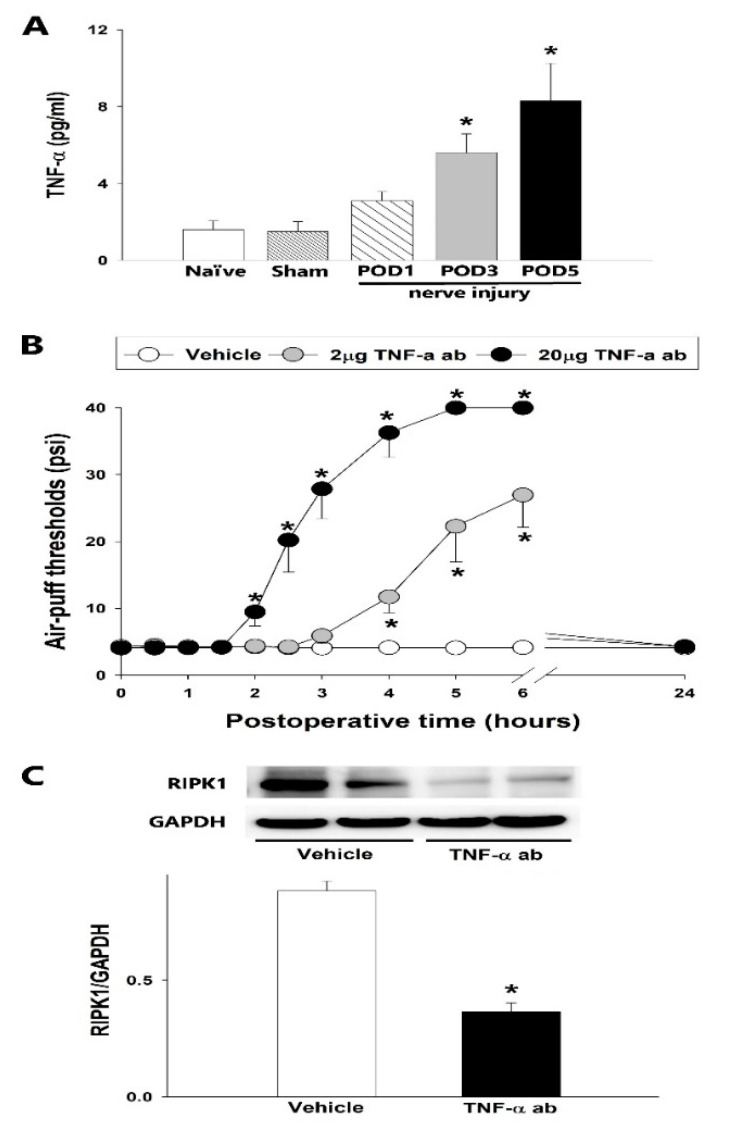
Effects of TNF-α antibody (ab) treatment of the trigeminal subnucleus caudalis upon mechanical allodynia and RIPK1 expression. (**A**) Time-course analysis of the changes in the TNF-α concentration after an inferior alveolar nerve injury produced by a malpositioned dental implant. ELISA analysis revealed significant increases in TNF-α concentration (one-way ANOVA with Holm–Sidak post hoc tests, *n* = 8) in the sham vs. inferior alveolar nerve injury POD 1, 3, and 5. (**B**) The intracisternal administration of TNF-α antibodies increased the air-puff thresholds compared with the vehicle-treated group (repeated measure ANOVA with Holm–Sidak post hoc tests, *n* = 7). Vehicle vs. 2 or 20 µg TNF-α-ab-treated group. (**C**) Western blot analysis revealed that RIPK1 expression was downregulated at 6 h after an intracisternal treatment with 20 µg TNF-α antibody on POD 3 compared with the vehicle group (Student’s *t*-test, *n* = 8). Vehicle vs. TNF-α-antibody-treated group. GAPDH was used as a loading control. All mean ± SEM, * *p* < 0.05.

**Figure 5 ijms-23-00506-f005:**
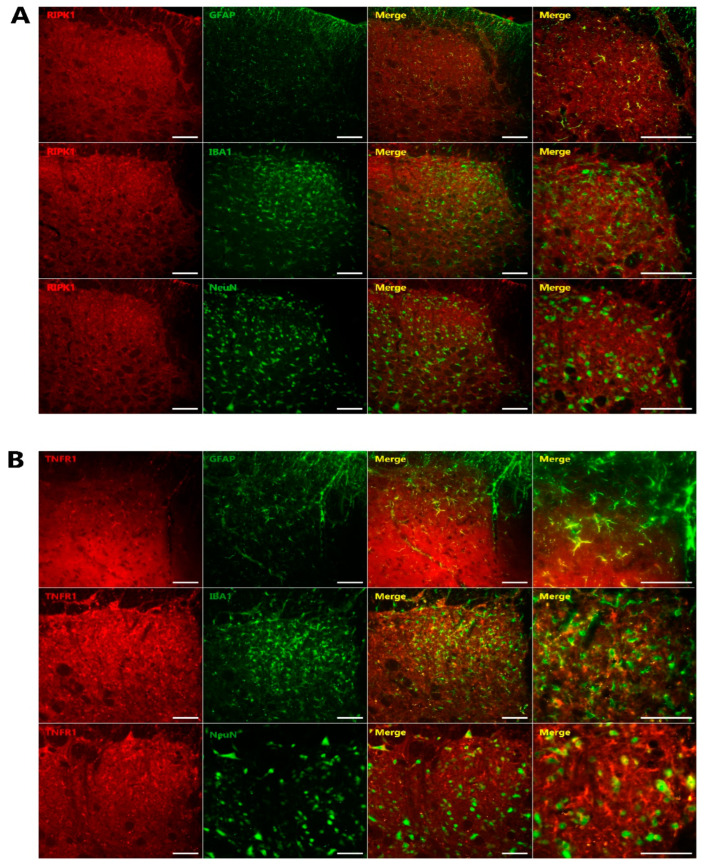
Characterization of RIPK1 and TNFR1 immunoreactive cells in the TSC after inferior alveolar nerve injury. (**A**) Double immunofluorescence analysis for RIPK1 (red) and NeuN, a neuronal marker (green); GFAP, an astrocyte marker (green); or IBA1, a microglia marker (green). RIPK1 immunoreactive cells were found to be mainly colocalized with GFAP, an astrocytic marker (green) (*n* = 6). (**B**) Double immunofluorescence staining for TNFR1 (red) with NeuN, GFAP, or IBA1 on POD 3. TNFR1 showed colocalization with GFAP (*n* = 6). Scale bars, 50 µm.

## Data Availability

Data are contained within the article.

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
