# Peer review of "TNF-α-Mediated RIPK1 Pathway Participates in the Development of Trigeminal Neuropathic Pain in Rats"

_ijms, 2022, doi:10.3390/ijms23010506_

Round 1

Reviewer 1 Report

The manuscript from the laboratory of an experienced and well-published investigator in the field of trigeminal pain mechanisms provides important, timely and new information related to the role of TNFα and RIPK 1 in mediating trigeminal neuropathic pain. Investigators provide evidence that the inferior alveolar nerve injury-induced mechanical allodynia is mediated by TNFα activated RIPK1 pathway. They showed quite clearly that nerve injury-induced allodynia is accompanied by upregulation of RIPK1 in the trigeminal nucleus caudalis, and intracisternal administration of an inhibitor of RIPK1 reversed the allodynia. Further, TNFα produced mechanical allodynia which was accompanied by upregulation of RIPK1, and the RIPK1 inhibitor reversed the TNFα – induced mechanical allodynia. TNFα and RIPK1 were shown to be co-localized in astrocytes. Experiments are designed very well, and data are analyzed and discussed appropriately.

Major Concerns: None

Minor concerns:

Results:

P 2, l74: delete “was” and replace basal value with basal levels

P3, l 81: delete was (Fig 1 legend)

P4, l96: at “a lower” before magnitude

P4: Figure 2. Please indicate in the figures: POD 3, POD 7 and POD-21

P7, l130: TNFα -induced mechanical allodynia starts at 30 minutes whereas RIPK1 is upregulated at 2 hours. Was the 30 minutes not included for analysis of RIPK1?

P8, Fig 5: Provide a bit more detail in the results and figure legend to guide the readers as to what they should be looking at in the merged images.

Discussion

P9, l173: delete the “comma” and “demonstrate”, add “that” before astroglial

P9, l174: add “to” before the development

P9, l179: delete “in the model rats”

P9, l181: replace with an astrocyte marker with “in astrocytes”

P9, l182: delete had been with “is; replace mediated with induced

P9, l189: replace anti-allodynic with allodynic

P9, l203: replace elicited with elicit

P9, l204: replace attenuate with produce

P9, l205-207: Rewrite the sentence. TNFα produces and not reduce pain behavior …..

P9, l213: delete associated with pain cognition

P9, l215: add “the” before development

P10, l229: add activation of before RIPK1

P19, l233: space between activates and RIPK1.

P10, l233; add on before our…

P10, l236: add the before development

Methods

P10, l240: adult and not adults

P10, l257: add from before animals

Author Response

Major Concerns: None

Results:

P2, l74: delete “was” and replace basal value with basal levels

P3, l81: delete was (Fig 1 legend)

P4, l96: at “a lower” before magnitude

: We made the appropriate corrections in the Results (line #74, page 2; line #81, page 3; line  #96, page 4).

P4: Figure 2. Please indicate in the figures: POD 3, POD 7 and POD-21

: We inserted POD3, POD 7, POD 21 in figure 2.

P7, l130: TNFα -induced mechanical allodynia starts at 30 minutes whereas RIPK1 is upregulated at 2 hours. Was the 30 minutes not included for analysis of RIPK1?

: We performed to evaluate changes in RIPK1 expression when TNF-α-induced mechanical allodynia was completely established. So, we examined RIPK1 expression 2 hours after intracisternal administration of rrTNF-α. Therefore, we did not examine changes in RIPK1 expression at this time point.

P8, Fig 5: Provide a bit more detail in the results and figure legend to guide the readers as to what they should be looking at in the merged images.

: We made the appropriate corrections in the Fig 5.

Discussion

P9, l173: delete the “comma” and “demonstrate”, add “that” before astroglial

P9, l174: add “to” before the development

: We made the appropriate corrections in the Discussions (line #173-174, page 9).

P9, l179: delete “in the model rats”

: We made the appropriate corrections in the Discussions (line #180, page 9).

P9, l181: replace with an astrocyte marker with “in astrocytes”

: We made the appropriate corrections in the Discussions (line #181-182, page 9).

P9, l182: delete had been with “is; replace mediated with induced

: We made the appropriate corrections in the Discussions (line #184, page 9).

P9, l189: replace anti-allodynic with allodynic

: We made the appropriate corrections in the Discussions (line #192, page 9).

P9, l203: replace elicited with elicit

: We made the appropriate corrections in the Discussions (line #206, page 9).

P9, l204: replace attenuate with produce

: We made the appropriate corrections in the Discussions (line #207, page 9).

P9, l205-207: Rewrite the sentence. TNFα produces and not reduce pain behavior …..

: We made the appropriate corrections in the Discussions (line #208-209, page 9).

P9, l213: delete associated with pain cognition

: We made the appropriate corrections in the Discussions (line #216-217, page 9).

P9, l215: add “the” before development

: We made the appropriate corrections in the Discussions (line #219, page 9).

P10, l229: add activation of before RIPK1

: We made the appropriate corrections in the Discussions (line #233, page 10).

P19, l233: space between activates and RIPK1.

: We made the appropriate corrections in the Discussions (line #239, page 10).

P10, l233; add on before our…

: We made the appropriate corrections in the Discussions (line #239, page 10).

P10, l236: add the before development

: We made the appropriate corrections in the Discussions (line #242, page 10).

Methods

P10, l240:adult and not adults

: We made the appropriate corrections in the Discussions (line #247, page 10).

P10, l257: add from before animals

: We made the appropriate corrections in the Methods (line #264, page 10).

Reviewer 2 Report

Young Son and colleagues in this study investigated the role of RIPK1 in the development of trigeminal neuropathic pain in rats. The analysis demonstrated that the nerve injury is correlated with an increase of the pro-inflammatory cytokine TNF-a and  the RIPK1 expression in the TSC. In particular the authors reported a co-localization of TNF receptor and RIPK1 with the GFAP marker identify the central role of astrocytes in the development of neuropathic pain. The work is well presented however some clarifications are needed.

-After injury the animals displayed a mechanical allodynia from day 1 till day 40, however the RIPK1 expression is statistically increase only on days 3 - 7. How can you explain this result?

-The doses of Nec-1 used should be justified

-The Nec-1 intracisternal injection demonstrated anti-allodynic effect on neuropathic animals however has been analyzed the effect of the treatment on naive rats? This data should be reported in the study. 

-The study demonstrated the therapeutic effect of Nec-1, when neuropathy was well established. Have the authors informations about a pre-treatment with Nec-1 before surgery on the development of trigeminal pain?

-The discussion needs to be improved by analyzing the data on the basis of the evidence in the literature

-The english should be improved

Author Response

Responses to Reviewer #2:

Young Son and colleagues in this study investigated the role of RIPK1 in the development of trigeminal neuropathic pain in rats. The analysis demonstrated that the nerve injury is correlated with an increase of the pro-inflammatory cytokine TNF-a and the RIPK1 expression in the TSC. In particular the authors reported a co-localization of TNF receptor and RIPK1 with the GFAP marker identify the central role of astrocytes in the development of neuropathic pain. The work is well presented however some clarifications are needed.

After injury the animals displayed a mechanical allodynia from day 1 till day 40, however the RIPK1 expression is statistically increase only on days 3 - 7. How can you explain this result?

: This present data showed that inferior alveolar nerve injury produced significant mechanical allodynia until POD 40. Also inferior alveolar nerve injury increased significantly RIPK1 expression in the trigeminal subnucleus caudalis on POD 3 and 7.  On POD 21, RIPK1 expression was also upregulated  by nerve injury from 0.54 ± 0.08 to 0.81 ± 0.124  in Fig 1, although it did not show statistical significant. Moreover, intracisternal administration of Nec-1, a RIPK1 Inhibitor, blocked mechanical allodynia produced by nerve injury on POD 21. These result suggest that upregulated RIPK1 expression may participate in the development of trigeminal neuropathic pain following inferior alveolar nerve injury.

-The doses of Nec-1 used should be justified

:  The previous study demonstrated that intrathecal injection of 1 or 10 μg of Nec-1 produced protective effects including decrease in spinal cord lesions and reducing ischemic and necrosis areas following traumatic spinal cord injury [1]. The present study demonstrated that intracisternal injection of Nec-1 at our doses (1 or 10 μg) attenuated mechanical allodynia in rats with inferior alveolar nerve injury.

: We inserted this information in the Methods (line #347-348, Page 12).

-The Nec-1 intracisternal injection demonstrated anti-allodynic effect on neuropathic animals however has been analyzed the effect of the treatment on naive rats? This data should be reported in the study.

à  We examined effects of Nec-1 on air-puff thresholds. The intracisternal administration of Nec-1 did not induce the mechanical allodynia in naive rats.

: We inserted this information in the Discussion (line #225-226, Page 10).

-The study demonstrated the therapeutic effect of Nec-1, when neuropathy was well established. Have the authors information about a pre-treatment with Nec-1 before surgery on the development of trigeminal pain?

: The reviewer's opinion shows a very important point. However, we did not perform pre-treatment with Nec-1 before surgery. We will try this experiments later.

-The discussion needs to be improved by analyzing the data on the basis of the evidence in the literature

: Several relevant evidences are added in Discussion section. (line #187-189, page 9; line #237-238, page 10).

-The English should be improved

: The manuscript was extensively corrected by a native speaker.

References

  1. Wang Y, Wang H, Tao Y, Zhang S, Wang J, Feng X. Necroptosis inhibitor necrostatin-1 promotes cell protection and physiological function in traumatic spinal cord injury. Neuroscience. 2014, 266, 91-101.

Round 2

Reviewer 2 Report

The authors provided an adequate modification to the main manuscript